# Pertussis Epidemiology in Children: The Role of Maternal Immunization

**DOI:** 10.3390/vaccines12091030

**Published:** 2024-09-09

**Authors:** Nicola Principi, Sonia Bianchini, Susanna Esposito

**Affiliations:** 1Università degli Studi di Milano, 20122 Milan, Italy; nicola.principi@unimi.it; 2Pediatric Unit, ASST Santi Carlo e Paolo, 20153 Milan, Italy; bianchini.sonia@outlook.it; 3Pediatric Clinic, Department of Medicine and Surgery, University of Parma, 43121 Parma, Italy

**Keywords:** *Bordetella pertussis*, maternal immunization, pertussis, pertussis vaccine, whooping cough

## Abstract

In the last twelve months, a significant global increase in pertussis cases has been observed, particularly among infants under three months of age. This age group is at the highest risk for severe disease, hospitalization, and death. Maternal immunization with the Tdap vaccine during pregnancy has been recommended to protect newborns by transferring maternal antibodies transplacentally. This review examines the current epidemiology of pertussis, the importance of preventing it in young children, and the effectiveness of maternal immunization. Despite the proven benefits of maternal vaccination, which has been found effective in pertussis prevention in up to 90% of cases, coverage remains suboptimal in many countries. Factors contributing to low vaccination rates include vaccine hesitancy due to low trust in health authority assessments, safety concerns, practical barriers to vaccine access, and the impact of the COVID-19 pandemic, which disrupted routine vaccination services. The recent increase in pertussis cases may also be influenced by the natural cyclic nature of the disease, increased *Bordetella pertussis* (*Bp*) activity in older children and adults, and the genetic divergence of circulating *Bp* strains from vaccine antigens. Given the high efficacy of maternal vaccination in preventing pertussis in infants, increasing coverage rates is crucial. Efforts to improve vaccine uptake should address barriers to access and vaccine hesitancy, ensuring consistent immune protection for the youngest and most vulnerable populations. Enhanced maternal vaccination could significantly reduce the incidence of whooping cough in infants, decreasing related hospitalizations and deaths.

## 1. Introduction

Over the past twelve months, there has been a notable and concerning increase in pertussis (whooping cough) cases across multiple countries, including regions in India and Africa. This rise in reported cases, while alarming, must be interpreted with caution due to several factors, such as incomplete case detection and variations in diagnostic methods across regions [1]. Nonetheless, the recent surge in diagnosed cases is difficult to dismiss, particularly given the scale of the increase and its widespread nature. For instance, in the United States, the number of pertussis cases reported in the early months of 2024 was significantly higher than during the same period in 2023. While these numbers are comparable to those observed in 2019, prior to the COVID-19 pandemic, they suggest that the current increase may not necessarily indicate a new pertussis epidemic. Instead, it might reflect a return to pre-pandemic pertussis patterns following the lifting of COVID-19 mitigation measures [2]. However, in many other countries, the epidemiological data suggest a different scenario, supporting the hypothesis of an actual pertussis epidemic.

In the European Union and European Economic Area (EU/EEA) countries, for example, there has been a sharp rise in pertussis cases since mid-2023, with 32,037 cases reported in just the first three months of 2024. This figure is comparable to the totals reported for the entire year in 2019 and in earlier years, indicating the possibility of an ongoing epidemic [3]. Similarly, in England, the situation has escalated dramatically. Between January and June 2024, the UK Health Security Agency (UKHSA) reported 10,493 laboratory-confirmed cases of pertussis, resulting in nine deaths. This represents a significant increase compared to the mere 856 cases reported throughout the entire year of 2023. Notably, the number of confirmed cases during the second quarter of 2024 surpassed those recorded in any quarter during the 2012 outbreak year, which was one of the most severe in recent history [4].

The rise in pertussis cases is not confined to high-income countries. In India, the situation has also shown troubling signs. According to recent data from the National Centre for Disease Control (NCDC) in India, there has been a 30% increase in pertussis cases in 2024 compared to the previous year, with approximately 7500 cases reported by mid-year [5]. This rise is particularly concerning in rural regions, where access to healthcare and vaccination is limited. The state of Uttar Pradesh alone reported over 1200 cases in the first half of 2024, nearly double the number reported in the same period of 2023 [5]. These figures highlight the urgent need for enhanced surveillance and vaccination efforts in India.

In Africa, the situation is equally alarming. Several countries have reported significant increases in pertussis cases. For instance, in Nigeria, the Ministry of Health recorded over 8000 pertussis cases between January and June 2024, marking a 40% increase compared to the same period in 2023 [6]. This rise is particularly notable in areas with low vaccination coverage, such as the northern regions of the country, where healthcare infrastructure is underdeveloped. Similarly, in South Africa, there has been a marked increase in pertussis incidence, with over 5000 cases reported in the first half of 2024, compared to 3200 cases in the same period of 2023 [6]. These increases have raised concerns about the effectiveness of existing vaccination programs and the need for more robust public health interventions.

Figure 1 summarizes the global trends in pertussis cases among infants under 3 months across different regions between 2023 and 2024.

The surge in pertussis cases has impacted all age groups, with the highest incidence observed among younger infants. In England, approximately 4% of these cases occurred in children under 3 months old, a rate that is double that of 2012, which was the last year before the introduction of preventive measures specifically targeting this vulnerable age group [4]. This age group is also at the greatest risk of mortality; alarmingly, 9 out of the 29 pertussis-related deaths between 2013 and 2024 occurred within the first six months of 2024 alone [4].

Maternal vaccination has been widely regarded as the most effective strategy for protecting neonates and young infants from pertussis before they are old enough to receive the primary immunization series themselves [7]. However, the global rise in cases among infants under 3 months old has raised important questions about the true effectiveness of this preventive measure and the potential influence of other external factors that might be contributing to the observed trends.

In this review, we delve into the current epidemiology of pertussis, emphasizing the critical importance of preventing this disease in young children. We also evaluate the effectiveness of maternal immunization with the Tdap vaccine, explore maternal immunization coverage rates, and examine the potential reasons behind the increased incidence of pertussis in younger children. To ensure a comprehensive and systematic review of the available literature on pertussis, we conducted an extensive search of the MEDLINE/PubMed database, covering the period from January 2000 to 15 May 2024. This time frame was chosen to capture the most relevant studies, particularly those reflecting changes in pertussis epidemiology, advances in vaccine development, and shifts in public health policies following the COVID-19 pandemic. The search strategy employed a combination of Medical Subject Headings (MeSH) terms and free-text keywords to maximize the retrieval of pertinent articles. The search terms included disease and pathogen-specific terms such as “pertussis”, “*Bordetella pertussis*”, and “whooping cough”. To address prevention and vaccination, we included terms like “pertussis prevention”, “pertussis vaccine”, and “Tdap vaccine”. Finally, to ensure relevance to the target populations, we used terms such as “children”, “infants”, and “maternal immunization”. Boolean operators were used to effectively combine these terms, ensuring that the search captured studies relevant to the key aspects of pertussis epidemiology, prevention, and vaccination. Articles were included in the review based on several criteria. We considered only peer-reviewed studies with rigorous study designs, including randomized placebo-controlled trials, controlled clinical trials, double-blind randomized controlled studies, systematic reviews, and meta-analyses. The review focused on studies involving children, infants, and pregnant women, as these populations are most relevant to the objectives of the review. Articles needed to report on outcomes related to pertussis incidence, the effectiveness of maternal immunization with the Tdap vaccine, vaccination coverage rates, or factors contributing to the increased incidence of pertussis in young children. Only articles published in English were included to maintain consistency, and the time frame was limited to studies published between January 2000 and 15 May 2024 to capture recent developments and long-term trends. We excluded articles that did not meet these criteria. Non-peer-reviewed articles, such as abstracts, case reports, commentaries, and editorials, were excluded due to their lower level of evidence. Studies focusing exclusively on adults or elderly populations, without addressing pertussis in children, infants, or pregnant women, were also excluded. Articles that did not report on relevant outcomes, such as those focused on unrelated respiratory illnesses or general public health interventions not specific to pertussis, were removed from consideration. Additionally, if multiple publications reported on the same study population or dataset, only the most comprehensive or recent publication was included to avoid duplication of data. The initial search identified a total of 1243 articles. After removing duplicates, 987 articles remained. We then screened the titles and abstracts of these articles for relevance, which led to the exclusion of 742 articles that did not meet the inclusion criteria. A detailed review of the full texts of the remaining 245 articles resulted in the exclusion of an additional 150 articles based on the aforementioned criteria. In the end, 70 articles were included in this review. These studies provided comprehensive data on the epidemiology of pertussis, the effectiveness of maternal immunization, vaccination coverage rates, and potential reasons behind the increased incidence of pertussis in young children. The included studies were critically appraised for quality and relevance, with a particular focus on those that provided robust data and clear conclusions. The findings from these studies form the basis of our discussion on the current state of pertussis prevention and the challenges that remain in protecting young children from this potentially deadly disease.

## 2. Why Pertussis Should Be Prevented in Younger Children

In infants, particularly those under 3 months of age, pertussis is an underdiagnosed, severe, and risky disease [8,9]. Underdiagnosis is due to the low sensitivity of clinical suspicion in these subjects. Unlike older children, younger infants do not exhibit the classic three stages of disease progression (catarrhal, paroxysmal, and convalescent), making diagnosis more challenging. Initial symptoms often resemble a mild, viral upper respiratory tract infection, including nasal congestion, runny nose, sneezing, mild or no fever, and watery eyes. As the disease progresses, cough frequency and severity increase, but rarely to paroxysmal levels. Respiratory symptoms can mimic bronchiolitis.

A study in Italy involving 195 children hospitalized for respiratory infection who were later found positive for *Bordetella pertussis* (*Bp*) infection revealed that pertussis was clinically suspected in only 68 cases (34.87%). The suspicion rate was significantly lower (27.9%) in children under 3 months, who were often misdiagnosed with bronchiolitis [10]. The most severe cases occur in children under 3 months. In the same study, it was noted that patients admitted to pediatric intensive care units (PICU) were younger than ward patients (42.8 vs. 240 days; *p* < 0.0007) and had longer hospital stays (24.7 vs. 7.52 days; *p* < 0.003). Severe manifestations can include apnea, pulmonary hypertension, acute respiratory distress syndrome, encephalopathy, respiratory failure, cardiovascular collapse, and septic shock [11]. Moreover, compared to mild cases, severe cases have a higher leukocyte count (35,800 ± 20,530/mm^3^ vs. 19,410 ± 8590/mm^3^) and severe hyperleukocytosis (18.18% vs. 0%, *p* < 0.05) [12]. Despite intensive care, death can occur. A study of 144 hospitalized children with severe pertussis, 56.9% of whom were under 3 months old, found that 38 patients were admitted to the PICU, and 13 died. Most deaths (77%) occurred in children under 6 weeks, with pulmonary hypertension (PHT) being the most common cause (odds ratio [OR] 323.29; 95% confidence interval [CI] 16.01–6529.42; *p* < 0.001) [13]. Similar findings were evidenced in a study carried out in eight French PICU enrolling 23 younger infants with severe pertussis. A total of 9 out of 23 (40%) died; they presented more frequently with cardiovascular failure (100% vs. 36%, *p* = 0.003) and PHT (100% vs. 29%, *p* = 0.002) than the survivors [14].

## 3. Maternal Vaccination for the Prevention of Pertussis in Younger Infants

Evidence indicating that younger infants are at the highest risk of severe pertussis, hospitalization, and death has led experts to recommend maternal immunization during the latter part of pregnancy. This strategy aims to reduce the clinical burden of pertussis in infants infected by *Bp* [15]. The transplacental transfer of maternal antibodies to the fetus is thought to protect the child after birth, especially during the first few months of life. On the other hand, administration of the pertussis vaccine to neonates and younger infants is not feasible due to their immature immune systems. Moreover, interventions such as the administration of tetanus toxoid, reduced diphtheria toxoid, and acellular pertussis vaccine (Tdap) to unvaccinated postpartum mothers and other family members of newborn infants to protect infants from pertussis (cocooning strategy) have proven difficult to implement and are generally ineffective [16]. Despite parents being considered putative transmitters of *Bp* to their infants, only few studies have shown a significant reduction of pertussis incidence in younger children whose parents had been immunized before or immediately after birth [17]. In contrast, several other studies have clearly shown that rates and severity of pertussis infection in younger infants did not differ after the implementation of postpartum cocooning [18,19,20,21,22].

Studies evaluating the impact of Tdap administration to pregnant women have consistently demonstrated its positive effects. Significant amounts of antibodies against vaccine antigens—pertussis toxin (PT), pertactin (PRN), and filamentous hemagglutinin (FHA)—have been detected in cord blood and young infant serum samples. Comparing pertussis antibody levels in cord blood of pertussis-vaccinated mothers to cord blood of control (placebo or unvaccinated) mothers, it was shown that geometric mean concentration (GMC) ratios (Tdap/control) for pertussis antibodies ranged from 2.7 to 22.2 for PT, 3.4 to 21.2 for FHA and 5.5 to 44.0 for PRN [23]. Unfortunately, a direct correlation of protection for pertussis is lacking. Moreover, comparisons between studies are difficult due to different laboratories and assays used, the timing of the vaccination during pregnancy, study design, and the epidemiological background of the study population. However, these antibody levels are considered indicative of substantial protection against the disease [16]. Clinical studies have confirmed this assumption [24,25]. For example, Amirthalingam et al. [25] compared pertussis epidemiology in England before and after the implementation of maternal immunization programs. With a coverage rate of 64%, maternal immunization resulted in a 78% (95% CI 72–83%) reduction in pertussis cases and a 68% (95% CI 61–74%) reduction in hospital admissions for children under 3 months. This age group was the only one where pertussis cases decreased compared to the pre-immunization period [25]. The evidence that Tdap can be administered to pregnant women without significant risk to the mother, fetus, or infant has further supported the introduction of maternal immunization in several countries [26].

Later studies have definitively confirmed the efficacy and clinical relevance of maternal immunization, addressing issues such as the optimal timing for Tdap administration and potential reactogenicity. An epidemiological evaluation conducted in England concluded that maternal immunization is effective even during periods of sustained *Bp* circulation and is independent of the acellular vaccine antigen composition [27]. The protection afforded by maternal immunization is extended by the administration of the infant’s first immunization dose. This study evaluated pertussis incidence across all ages during the three years following the introduction of maternal immunization, a period marked by a significant increase in pertussis cases among those over one year old. Compared to the three years before the maternal program’s introduction, cases in infants under three months dropped from 60.4 to 12.7 per 100,000, with vaccine efficacy over 90%. Pertussis-related deaths also decreased, with efficacy calculated at 95% (95% CI 79–100%) [27]. Additionally, maternal vaccination was found effective even after infants received their first primary dose (VE, 82%; 95% CI 65–91%).

In the USA, a retrospective cohort study enrolling 148,981 neonates born in California from 2010 to 2015 has shown [28] that the efficacy of maternal Tdap administration was 91.4% (95% CI; 19.5–99.1%) during the first 2 months of life and 69.0% (95% CI 43.6–82.9%) during the entire first year of life. Similar findings at least for children in the first two months of life were reported in a time-trend analysis of infant pertussis from 1 January 2000 to 31 December 2019 [29]. Comparing the pre-maternal Tdap vaccination period (2000–2010) with the post-maternal Tdap vaccination period (2012–2019) and focusing on infants younger than 2 months and those aged 6–11 months, the impact of maternal vaccination on pertussis incidence was measured by evaluating slope differences between the two periods. During the study period, 57,460 pertussis cases in children under 12 months were reported, with 19,322 (33.6%) in infants under 2 months. Before Tdap vaccine administration to pregnant women, annual pertussis incidence did not change in children under 2 months and slightly increased in older infants. However, after introducing maternal vaccination, a significant reduction in pertussis incidence in infants under 2 months was observed (slope, −14.53 per 100,000 infants per year; *p* = 0.001), whereas incidence in the older group did not change significantly (slope, 1.42 per 100,000 infants per year; *p* = 0.29) [30].

Initially, maternal immunization was recommended at 28–32 weeks of gestation to balance antibody transplacental passage and pertussis IgG level decay in fetal blood, ensuring elevated antibody levels in cord blood. However, the optimal timing for Tdap administration during pregnancy remains debated, with studies suggesting different recommendations. Abu Raya et al. reported that both PT and FHA concentrations were significantly higher in newborns’ cord sera when immunization occurred during gestational weeks (GW) 27–30 (+6) compared to GW 31–36 and GW > 36 [30]. Eberhart et al. found higher geometric mean concentrations (GMCs) of cord blood antibodies to PT and FHA in children born full-term to mothers who received Tdap in the second trimester (GW 13–25) compared to those immunized in the third trimester (GW ≥ GW) [31].

In the UK, it was found that the efficacy of maternal vaccination was equivalent in infants with mothers vaccinated in the second or in the third trimester of pregnancy [32]. On the contrary, data collected in the USA seemed to suggest that vaccination during the third trimester could be more effective (77.7%; 95% CI, 48.3–90.4) than first- or second-trimester vaccination (64.3%; 95% CI −13.8% to 88.8%), although CIs overlapped [33]. This explains why scientific societies have revised their recommendations, expanding the gestational window for maternal immunization, despite the slight differences regarding the best time for vaccine administration. In the EU/EEA, Tdap vaccination is recommended between GA 16 and 36 weeks [34], in the UK from GA 16 to 32 weeks, with a preference for around GA 20 weeks, and in the USA, vaccination is recommended between GA 27 and 36 weeks, preferably early in this period [35]. However, the flexibility supports earlier maternal immunization, which may protect preterm neonates and increase vaccination opportunities.

Table 1 shows the key findings related to the effectiveness of maternal Tdap immunization based on various studies selected in the manuscript.

Concerns that maternal immunization might erode the effectiveness of primary vaccination at later ages appear minimal. Studies have shown that children born to vaccinated mothers have a lower immune response to primary pertussis immunization than those without maternal immunization, with reduced antibody concentrations against several pertussis antigens [36,37] and lower antibody avidity [38]. Moreover, it was found that the blunting response following maternal pertussis immunization is heterologous, also causing decreased immune response to the polio vaccine and to other vaccines that contain modified diphtheria or tetanus toxins as carrier proteins, such as pneumococcal conjugate vaccines [39]. Despite these findings, pertussis incidence evaluations in children with and without maternal immunization who received recommended pertussis vaccine doses during infancy suggest that any risk is minimal and can be masked by natural variations in pertussis incidence over time. Briga et al. concluded that no reason exists to debate maternal immunization as a mandatory measure to reduce pertussis risks in younger infants [40].

## 4. Maternal Immunization Coverage

The evidence that maternal Tdap immunization reduces infant pertussis risk quickly led to official national recommendations in the USA [41] and the UK [42] in 2011 and 2012, respectively. In subsequent years, maternal immunization was recommended in several other countries, including 24 of the 30 EU/EEA countries, Canada, Australia, and most Central and South American countries [7,43]. However, many populous and developed countries, such as China and Japan [44,45], and EU/EEA countries like Bulgaria, Estonia, Finland, Malta, and Slovakia, have not yet implemented national Tdap administration initiatives for pregnant women [46].

Even in countries with strong maternal immunization support, coverage remains suboptimal. In the USA, six years after the Advisory Committee on Immunization Practices (ACIP) recommended maternal immunization, coverage was only 56.3% and 31.4% in 2017, based on data from the MarketScan Commercial and Multi-State Medicaid Databases [47]. These levels remained unchanged before the COVID-19 pandemic [48] and when the pandemic was nearing its end. During the 2022–2023 influenza season, coverage was only 55.4% [49].

In England, between 1 October 2012 and 3 September 2013, the average vaccine coverage was 64% [25]. The highest coverage was 76% in December 2016, but it gradually declined to 58% in June 2023 [50]. In the EU/EEA, only nine countries reported maternal immunization coverage in 2023, with significant variation from 1.6% in Czechia to 88.5% in Spain. Coverage in Slovenia, Romania, and Germany was 6.5%, 8.8%, and 39.7%, respectively [46]. Some countries also reported lower coverage in 2023 than in previous years, indicating a decline in adherence to official recommendations.

Figure 2 illustrates the relationship between maternal immunization coverage and pertussis incidence among infants under 3 months old. Each point represents a different country, showing how higher vaccination coverage generally correlates with lower incidence rates.

Several factors may explain the lower-than-expected maternal Tdap vaccination coverage. A systematic review and meta-analysis of studies published by 22 November 2018, on maternal vaccine acceptance found that healthcare professional (HCP) recommendation is the most important factor influencing uptake. When a specialist in obstetrics and gynecology or a midwife recommends the vaccine, the odds of accepting pertussis immunization increase tenfold (OR 10.33, 95% CI 5.49–19.43) [51]. However, other factors may lead pregnant women to disregard HCP recommendations. Over 30% of pregnant women refuse the Tdap vaccine despite HCP suggestions [52]. Low trust in health authority assessments has been linked to vaccine hesitancy and refusal [53]. A 2019 study in Norway involving 1148 pregnant women at GA 20–40 weeks confirmed that this applies to Tdap maternal vaccinations [54].

Safety concerns or fear of side effects are significant factors, overshadowing the risk of severe disease in the child. Infertility, autism spectrum disorders, and pregnancy-related problems like miscarriage, preterm birth, and birth defects are common reasons for vaccine hesitancy [55,56,57]. Practical barriers, including physical accessibility, also impact immunization coverage, even among mothers initially positive about vaccination [58]. The COVID-19 pandemic further exacerbated these issues. A review of studies from 29 countries showed a general decline in vaccination coverage (up to −79%) during the pandemic, with significant disruptions in vaccination service accessibility and delivery [59]. This decline affected all vaccines, including those containing pertussis antigens [60].

## 5. Potential Reasons for the Increase in Pertussis Incidence in Younger Infants

Several factors could explain the recent significant increase in pertussis cases among infants under three months of age. Epidemiological studies have shown that larger epidemics of pertussis tend to occur every three to five years, even without significant variations in vaccination coverage [61]. This suggests that the current epidemic, which also affects younger infants, may be a manifestation of the natural cyclic nature of the disease, as suggested by US health authorities. It is well known that a pertussis vaccine administered in the first year of life provides immunity that declines over time and that most of adults do not have pertussis-specific antibodies and are vulnerable to infection of *Bp* [62,63]. Moreover, the acellular pertussis vaccine, currently used in most countries for immunization, prevents people from getting sick, but it does not prevent them from becoming infected and spreading the disease. Both factors create the basis for the development of periodic epidemics that can have the highest occurrence in children during the first months of life, especially those without any immune protection. The current pertussis epidemic has been observed across all age groups, indicating a greater exposure risk for younger infants. Studies have shown that parents (20–48%) and siblings (19–53%) are common sources of *Bp* infection for infants [64,65]. Thus, a sudden increase in *Bp* activity in the general population may have significantly contributed to the rise in pertussis cases in younger infants.

The increased circulation of *Bp* may be partly attributed to the restrictions implemented to contain the COVID-19 pandemic. During 2020–2022, pertussis prevalence was very low, which may have decreased natural boosting and increased the proportion of the population susceptible to pertussis. In the EU/EEA, the notification rate in 2022 was 0.4 cases per 100,000 population, which was the lowest rate in over a decade. [66,67]. In the USA, in 2021 only 2116 pertussis cases were reported, compared to 18,917 in 2019. When *Bp* began to circulate again, the number of pertussis cases increased. Additionally, the reduction in vaccine administration during the COVID-19 pandemic contributed to the increase in unprotected individuals. A study in 10 US jurisdictions during March–May 2020 showed that DTaP vaccine doses administered to children under 24 months and children aged 2–6 years declined by a median of 15.7% and 60.3%, respectively, compared with the same period in 2018 and 2019 [68].

Another potential factor in the recent pertussis epidemic could be the genetic divergence of circulating *Bp* strains away from vaccine antigens. Evidence suggests that the use of acellular pertussis vaccines has led to the emergence of *Bp* strains with increased PT production, PRN deficiency, and mutations in other vaccine antigens, potentially enabling vaccine escape. Currently, over 90% of Bp strains circulating in the EU possess one or more of these genotypes, which may contribute to the reduced efficacy of maternal vaccination in pregnant women and the increase in pertussis cases among younger infants [69].

Regarding maternal immunization, coverage has remained relatively unchanged in many countries and has slightly declined in others, with declining trends starting before the recent pandemic. This suggests that recent variations in maternal immunization coverage have played a minor role in the development of the epidemic in younger infants. More significant, however, is the generally low maternal coverage, which leaves many younger infants without consistent immune protection during the first weeks of life worldwide. This, combined with increased *Bp* activity, appears to explain most of the recent pertussis cases in younger infants [69]. Given the high effectiveness of maternal vaccination, higher coverage levels than those currently present in various countries could significantly reduce the number of whooping cough cases in younger infants, thereby reducing hospitalizations and deaths from this serious infectious disease [70].

## 6. Conclusions

In recent months, a pertussis epidemic has emerged globally, with a particularly alarming rise in cases among infants under three months old—the age group most vulnerable to hospitalization and death from the disease. The review highlights that maternal immunization is a critical strategy for protecting these young infants, significantly reducing their risk of contracting pertussis. Despite the well-documented benefits of maternal immunization, the current pertussis control strategies appear insufficient, as evidenced by the surge in cases.

Available data consistently show that maternal immunization provides significant protective effects against pertussis in newborns. However, the recent increase in whooping cough cases among infants under three months old suggests that the broader circulation of *Bp* in the general population has led to higher exposure risks for these vulnerable infants. This trend underscores a critical gap in current pertussis control strategies, namely low maternal vaccination coverage.

To effectively combat this epidemic, it is imperative to adopt a multifaceted approach to enhance pertussis prevention. First and foremost, increasing maternal vaccination coverage must be a priority. This requires concerted efforts at both policy and community levels. Health authorities should implement targeted campaigns to raise awareness about the importance of maternal immunization, particularly focusing on pregnant women and healthcare providers. These campaigns should address common barriers to vaccination, such as misinformation, vaccine hesitancy, and limited access to healthcare services.

In addition to boosting maternal immunization rates, broader public health strategies should be reinforced. These include strengthening routine childhood vaccination programs, ensuring timely administration of booster doses, and enhancing surveillance systems to monitor pertussis incidence more effectively. Furthermore, public health policies should consider the integration of pertussis vaccination into prenatal care routines, making it a standard practice during pregnancy. This could be supported by providing vaccines at no cost in prenatal clinics and offering incentives for healthcare providers to promote and administer the vaccine.

Another key recommendation is the expansion of research initiatives to better understand the epidemiology of pertussis and the factors contributing to the recent increase in cases. This includes investigating potential changes in the pathogen itself, such as mutations that may affect vaccine efficacy, and exploring the impact of population immunity levels following the COVID-19 pandemic. Research should also focus on optimizing vaccine formulations and schedules to enhance their effectiveness in different populations.

Finally, international collaboration is crucial in addressing this global health challenge. Countries should share data, strategies, and resources to create a coordinated response to the pertussis epidemic. Global health organizations can play a pivotal role in facilitating these collaborations and providing guidance on best practices for pertussis control.

## Figures and Tables

**Figure 1 vaccines-12-01030-f001:**
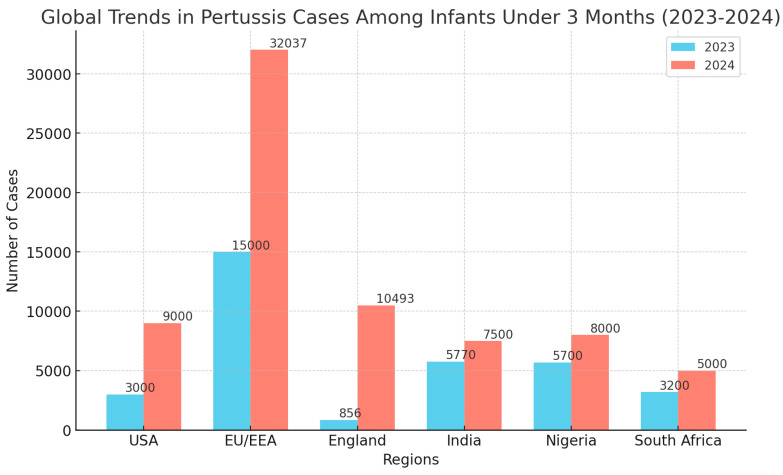
Global trends in pertussis cases among infants under three months of age (2023–2024).

**Figure 2 vaccines-12-01030-f002:**
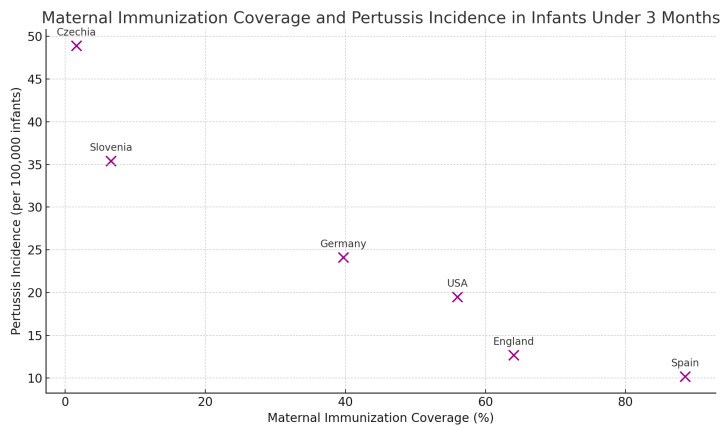
Maternal immunization coverage and pertussis incidence in infants under 3 months.

**Table 1 vaccines-12-01030-t001:** Effectiveness of maternal immunization against pertussis.

Study/Author	Country	Year	Key-Findings
Amirthalingam et al. [25]	England	2012–2013	Effectiveness of 78%
Baxter et al. [28]	USA	2010–2015	Effectiveness of 91.4%
Skoff et al. [33]	USA	2000–2019	Effectiveness of 77.7% (3rd trimester)
Eberhardt et al. [31]	Switzerland	2022	Higher antibody levels when administered in 2nd trimester
Perrett et al. [23]	Multiple countries	2020	High antibody transfer

## Data Availability

Not applicable.

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
