# Peer review of "Pertussis Epidemiology in Children: The Role of Maternal Immunization"

_vaccines, 2024, doi:10.3390/vaccines12091030_

Round 1

Reviewer 1 Report

Comments and Suggestions for Authors

This is a well written paper regarding the epidemiology of pertussis in children and the role of maternal immunisation.

The background is relevant and concise.

The methodology could be described more fully - how was the literature search conducted, a summary of articles included, which were excluded for what reason etc. 

The findings are well described and well organised. The review does not include any information from India and Africa although peer reviewed articles are available.

The language and sentence structure are used is easy to read.

Maybe adding a figure Illustrating search strategy would be helpful.

Author Response

This is a well written paper regarding the epidemiology of pertussis in children and the role of maternal immunisation.

Re: Thank you for your comments. We revised the manuscript according to your suggestions and those received from the other reviewers.

The background is relevant and concise.

Re: Thank you for your positive evaluation. We expanded the text according to the suggestions of the Editor (pp. 2-3).

The methodology could be described more fully - how was the literature search conducted, a summary of articles included, which were excluded for what reason etc. 

Re: We improved the text as suggested.

The findings are well described and well organised. The review does not include any information from India and Africa although peer reviewed articles are available.

Re: We improved the text adding data from India and Africa (pp. 2-3).

The language and sentence structure are used is easy to read.

Re: Thank you very much for the positive evaluation.

Maybe adding a figure Illustrating search strategy would be helpful.

Re: We reported details in the text.

Reviewer 2 Report

Comments and Suggestions for Authors

The authors provide an up-to-date and concise review on recent pertussis epidemiology and role of maternal immunization with TdaP on epidemiology of pertussis in infants <3 months of age.The manuscript is clear and presented in well structured manner. Recent data on epidemiology of pertussis in the Europe and USA are shown.The possible reasons for the increase of pertussis in recent years are presented. The vaccination of mothers during pregnancy against Bp demonstrated positive effects in clinical trials.  Due to low coverage of maternal  TdaP vaccinations in all countries the impact of  mothers  vaccination on Bp epidemiology in infants <3 months could not be shown. Abstract and conclusion are written well. The paper has the added value.

Author Response

The authors provide an up-to-date and concise review on recent pertussis epidemiology and role of maternal immunization with TdaP on epidemiology of pertussis in infants <3 months of age.The manuscript is clear and presented in well structured manner. Recent data on epidemiology of pertussis in the Europe and USA are shown.The possible reasons for the increase of pertussis in recent years are presented. The vaccination of mothers during pregnancy against Bp demonstrated positive effects in clinical trials.  Due to low coverage of maternal  TdaP vaccinations in all countries the impact of  mothers  vaccination on Bp epidemiology in infants <3 months could not be shown. Abstract and conclusion are written well. The paper has the added value.

Re: Thank you very much for your positive evaluation. We improved the text according to other reviewers’ suggestions.

Reviewer 3 Report

Comments and Suggestions for Authors

dear authors

Thank you for the opportunity to review the Literature review manuscript entitled “Pertussis Epidemiology in Children: The Role of Maternal Immunization” by Nicola Principi et al.

My decision is Accept after major revision.

The paper is interesting and deal with the emerging topic. The introduction is clear and well arranged. The discussion is good even could be improved. I think the major issue is the methodology of the manuscript. Authors should be describing their selection criteria for studies to be involved in this review.

General recommendations

According to ECDC during 2023-24, in 17 EU/EEA countries, infants (those under the age of one year) represented the group with the highest reported incidence, whereas in six countries, the highest incidence is reported in adolescents 10-19 years. The majority of deaths occurred in infants.

·          The authors should be discussed and other strategies to eliminate pertussis such a “cocooning strategy” in combination with low vaccination coverage to the elderly. Suggested references: MMWR Morb Mortal Wkly Rep. 2011;60(41):1424–1426.

·      Please discussed what is the optimal timing of Tdap vaccination during pregnancy to maximize protection of the newborn.

      Suggested references: https://doi.org/10.1080/14760584.2020.1791092

Abstract

Reform the affiliations according to Journal suggestions

Introduction

Lines 52-53: “Maternal vaccination is considered the most effective measure to prevent pertussis” Despite the overall risk is assessed as high for unimmunised or partially immunised infants<6 months of age, please reform the phrase, the most effective measure is the timeline and full completion of pertussis primary immunization of infants followed by other vaccinations strategies like maternal Immunization.

Line 75: report the source of the photo  

Lines 236-243: Please add and discussed the follow references to potential factor in the recent pertussis epidemic:

·             https://doi.org/10.1038/s41467-021-23114-y

·             https://doi.org/10.3390/vaccines10091511  

References

Please reform the numerical status of references according to Journal suggestions.

Author Response

Dear authors

Thank you for the opportunity to review the Literature review manuscript entitled “Pertussis Epidemiology in Children: The Role of Maternal Immunization” by Nicola Principi et al.

My decision is Accept after major revision.

The paper is interesting and deal with the emerging topic. The introduction is clear and well arranged. The discussion is good even could be improved. I think the major issue is the methodology of the manuscript. Authors should be describing their selection criteria for studies to be involved in this review.

Re: Thank you very much for your suggestions. We revised the text according to your comments and those received from the other reviewers.

General recommendations

According to ECDC during 2023-24, in 17 EU/EEA countries, infants (those under the age of one year) represented the group with the highest reported incidence, whereas in six countries, the highest incidence is reported in adolescents 10-19 years. The majority of deaths occurred in infants.

The authors should be discussed and other strategies to eliminate pertussis such a “cocooning strategy” in combination with low vaccination coverage to the elderly. Suggested references: MMWR Morb Mortal Wkly Rep. 2011;60(41):1424–1426.

Re: Details regarding cocooning strategy have been included together with some references.

Please discussed what is the optimal timing of Tdap vaccination during pregnancy to maximize protection of the newborn.

Suggested references: https://doi.org/10.1080/14760584.2020.1791092

Re: The optimal timing for maternal immunization has been discussed and the reference you mentioned was added.

Reform the affiliations according to Journal suggestions

Re: Affiliations have been reformed according to journal suggestions

Introduction

Lines 52-53: “Maternal vaccination is considered the most effective measure to prevent pertussis” Despite the overall risk is assessed as high for unimmunised or partially immunised infants<6 months of age, please reform the phrase, the most effective measure is the timeline and full completion of pertussis primary immunization of infants followed by other vaccinations strategies like maternal Immunization.

Re: It has been specified that maternal immunization is the best solution to prevent pertussis in younger infants before they receive the primary immunization.

Line 75: report the source of the photo 

Re: The photo has been deleted. It is an old photo and we did not find parents’ consent.

Lines 236-243: Please add and discussed the follow references to potential factor in the recent pertussis epidemic:  https://doi.org/10.1038/s41467-021-23114-y,            https://doi.org/10.3390/vaccines10091511 

Re: All the section regarding potential factors in the recent pertussis epidemic has been totally rewritten, according to the suggestions. The two references have been added.

References: Please reform the numerical status of references according to Journal suggestions.

Re: Done.

Reviewer 4 Report

Comments and Suggestions for Authors

Comments

This review explores the epidemiology of pertussis in children, particularly focusing on the effectiveness of maternal immunization with the Tdap vaccine and the recent increase in pertussis cases globally, especially among infants under three months of age. The review discusses current epidemiology, the importance of preventing pertussis in young children, maternal immunization coverage, and possible reasons for increased incidence. This review provides valuable insights into pertussis epidemiology and maternal immunization but requires more robust data support and improved structural organization.

Major Issues

1.        Logical Structure: The overall structure of the review appears somewhat scattered. It is recommended to reorganize the sections to follow a clearer logical sequence, such as “current status - issues - impact - recommendations”.

2.        Insufficient Data and Evidence: The review mentions the increase in pertussis cases but lacks detailed statistical data to support this claim. It is crucial to include specific epidemiological data and related charts for better illustration. For example, adding a figure in the Background section to show the increases in pertussis cases worldwide.

3.        Effectiveness of Maternal Immunization: The discussion on the effectiveness of the Tdap vaccine during pregnancy is relatively vague. More detailed data on vaccine efficacy and real-world case studies should be provided.

4.        Influence of External Factors: The review briefly mentions external factors affecting the incidence of pertussis. It is necessary to specify and elaborate on these factors.

5.        Weak Conclusion and Recommendations: The conclusion lacks depth in evaluating current pertussis control strategies and future recommendations. It is suggested to provide more detailed prevention strategies and policy directions.

6.        In Figure 1, a 3-month infant with pertussis was showed. The authors must pay attention to personal privacy and medical ethics, such as whether informed consent was sought from the infant's family, and if informed consent was obtained, the authors should have shielded the infant's face to avoid compromising personal privacy.

Minor Issues

1.        Language and Sentences: Some sentences are overly complex. Simplify the language to avoid long compound sentences and improve readability.

2.        Lack of Figures and Tables: Although multiple data points are mentioned, the review lacks relevant figures and tables. Including these would help in visually presenting the data and trends.

3.        Outdated References: Some references are outdated. Updating the reference list with the most recent research findings and data is recommended.

4.        Format Consistency: There are inconsistencies in the document formatting, such as heading levels and numbering styles. Ensure all formatting follows a consistent standard for professional presentation.

5.        Glossary of Acronyms: Terms like “Tdap” should be spelled out with their full form at the first mention for clarity.

6.        Maternal Immunization Coverage Data: The discussion on maternal immunization coverage being relatively unchanged needs specific data or reference to previous studies for substantiation.

7.        Citation Format: Several citations lack proper formatting or are inconsistent. Verify and standardize the citation format to meet journal requirements.

8.        Time Expression Consistency: Phrases like “recent months” and “recent years” show inconsistent time expressions. Use precise time frames for accurate communication.

9.        Incomplete Abstract: The abstract does not fully cover the contents of the review. Ensure the abstract briefly mentions key findings and conclusions for completeness.

In summary, while the review provides important insights into pertussis and maternal immunization, there is room for significant improvement in terms of structure, data support, and detail. The authors are encouraged to revise and enhance the manuscript based on these suggestions.

Comments on the Quality of English Language

See my comments

Author Response

This review explores the epidemiology of pertussis in children, particularly focusing on the effectiveness of maternal immunization with the Tdap vaccine and the recent increase in pertussis cases globally, especially among infants under three months of age. The review discusses current epidemiology, the importance of preventing pertussis in young children, maternal immunization coverage, and possible reasons for increased incidence. This review provides valuable insights into pertussis epidemiology and maternal immunization but requires more robust data support and improved structural organization.

Re: Thank you very much for your suggestions. We improved the text according to your comments and those received from the other reviewers.

Major Issues

  1. Logical Structure: The overall structure of the review appears somewhat scattered. It is recommended to reorganize the sections to follow a clearer logical sequence, such as “current status - issues - impact - recommendations”.

Re: The paper follows a logical structure as it discusses the present pertussis epidemic in younger children and the factors that could explain it. As maternal immunization is considered the best measure to reduce pertussis incidence in neonates and younger infants, true effectiveness of this measure and present coverage are detailed. Finally, the importance of poor coverage in causing present epidemic is detailed.

  1. Insufficient Data and Evidence: The review mentions the increase in pertussis cases but lacks detailed statistical data to support this claim. It is crucial to include specific epidemiological data and related charts for better illustration. For example, adding a figure in the Background section to show the increases in pertussis cases worldwide.

Re: The Background section has been expanded according to the Editor’s request. Data on pertussis incidence in India and Africa have been added. Moreover, a Figure has been added as suggested.

  1. Effectiveness of Maternal Immunization: The discussion on the effectiveness of the Tdap vaccine during pregnancy is relatively vague. More detailed data on vaccine efficacy and real-world case studies should be provided.

Re: Further data have been added.

  1. Influence of External Factors: The review briefly mentions external factors affecting the incidence of pertussis. It is necessary to specify and elaborate on these factors.

Re: Further details at this regard have been added.

  1. Weak Conclusion and Recommendations: The conclusion lacks depth in evaluating current pertussis control strategies and future recommendations. It is suggested to provide more detailed prevention strategies and policy directions.

Re: We improved the Conclusions section according to your suggestions.

  1. In Figure 1, a 3-month infant with pertussis was showed. The authors must pay attention to personal privacy and medical ethics, such as whether informed consent was sought from the infant's family, and if informed consent was obtained, the authors should have shielded the infant's face to avoid compromising personal privacy.

Re: The photo has been deleted. It is an old photo and we did not find parents’ consent.

Minor Issues

  1. Language and Sentences: Some sentences are overly complex. Simplify the language to avoid long compound sentences and improve readability.

Re: The text has been reviewed by an English mother tongue with appropriate knowledge on the topic of the manuscript.

  1. Lack of Figures and Tables: Although multiple data points are mentioned, the review lacks relevant figures and tables. Including these would help in visually presenting the data and trends.

Re: Two Figures and one Table has been added as suggested.

  1. Outdated References: Some references are outdated. Updating the reference list with the most recent research findings and data is recommended.

Re: According to another reviewer’s suggestions, we reported in details how references were selected. References have been updated and new references have been added.

  1. Format Consistency: There are inconsistencies in the document formatting, such as heading levels and numbering styles. Ensure all formatting follows a consistent standard for professional presentation.

Re: The text has been revised according to your comment.

  1. Glossary of Acronyms: Terms like “Tdap” should be spelled out with their full form at the first mention for clarity.

Re: Done.

  1. Maternal Immunization Coverage Data: The discussion on maternal immunization coverage being relatively unchanged needs specific data or reference to previous studies for substantiation.

Re: New references have been added.

  1. Citation Format: Several citations lack proper formatting or are inconsistent. Verify and standardize the citation format to meet journal requirements.

Re: Revised accordingly.

  1. Time Expression Consistency: Phrases like “recent months” and “recent years” show inconsistent time expressions. Use precise time frames for accurate communication.

Re: Revised has suggested.

  1. Incomplete Abstract: The abstract does not fully cover the contents of the review. Ensure the abstract briefly mentions key findings and conclusions for completeness.

Re: The abstract has been revised according to your suggestion, although we maintained it with the length recommended in the journal’s instructions for authors.

 In summary, while the review provides important insights into pertussis and maternal immunization, there is room for significant improvement in terms of structure, data support, and detail. The authors are encouraged to revise and enhance the manuscript based on these suggestions.

Re: The text has been improved according to your recommendations. We hope that you could accept the manuscript in its present form.

Round 2

Reviewer 3 Report

Comments and Suggestions for Authors

Dear authors thank you for thhe revisions.

Reviewer 4 Report

Comments and Suggestions for Authors

The authors have addressed all my concerns.